

# ASM-SS: The First Quasi-Global High Spatial Resolution Coastal Storm Surge Dataset Reconstructed from Tide Gauge Records

Lianjun Yang[1], Taoyong Jin[1,2], and Weiping Jiang[1,2]

[1]School of Geodesy and Geomatics, Wuhan University, Wuhan 430079, China;

5 [2]MOE Key Laboratory of Geospace Environment and Geodesy, Wuhan University, Wuhan 430079, China.

*Correspondence to*: Taoyong Jin (tyjin@sgg.whu.edu.cn)

**Abstract.** Storm surges (SSs) cause massive loss of life and property in coastal areas each year. High spatial resolution and long-term SS records are the basis for deepening our understanding of this disaster. However, such global or quasi-global scale information could only be simulated by global numerical models until now due to the sparse and uneven distribution of tide gauge stations. In this paper, the all-site modeling framework for the data-driven model was implemented on a quasi-global scale within areas severely affected by SSs caused by tropical and extratropical cyclones. Compared to single-site modeling data-driven models, it can provide SS information for ungauged points. Compared to numerical models, it can reconstruct long-term SSs faster with fewer computational resources. We generated the first high spatial resolution (every 10 km per station along the coastline) hourly SS dataset ASM-SS (all-site modeling storm surge) within 45°S to 45°N, whose record length is over 80 years from 1940 to 2020. Assessments indicate that for 95th extreme SSs, the precision of this model (medians of correlation coefficients, root mean square errors, and mean biases are 0.66, 9 cm, and -4.4 cm, respectively) is slightly better than that of the state-of-the-art global hydrodynamic model (medians are 0.58, 10.8 cm, and -4.3 cm); for annual maximum SSs, our model is more stable than the numerical model with overall root mean square error and coefficient of determination optimizing by around 23.1% and 14.8%, respectively. This dataset could provide possible alternative support for coastal communities to estimate return levels of extremes, analyze variations (intensity, frequency, and trend) of SSs, and other relevant applications. The ASM-SS dataset is available at https://doi.org/10.5281/zenodo.13293595 (Yang et al., 2024a).

## 1 Introduction

Extreme sea level (ESL) events, defined as exceptional variations of sea-surface height caused by tides, storm surges, and sea-surface waves (Gregory et al., 2019), lead to severe economic losses globally each year (Kron, 2013). Around 680 million people living in low-lying coastal zones with elevation lower than 10 m above sea level (Pörtner et al., 2022) are already directly or indirectly affected by ESLs in current climate conditions (Hinkel et al., 2014). Even more concerning, the impacts of ESLs are expected to intensify in the future due to the rise in global sea level (Palmer et al., 2021), the increasing intensity of tropical cyclones (Knutson et al., 2020), and the growth of coastal population (Merkens et al., 2016). Storm surges (SSs) caused by tropical and extratropical cyclones have significant uncertainty compared to deterministic and predictable tides.



Understanding how SSs varied in different regions, interacted with other components, and responded to climate change in the past can better prepare coastal communities for incoming ESLs.

  To date, SS analysis can be roughly divided into four categories: (1) evaluating return levels of extremes; (2) assessing the contribution of SSs to ESLs; (3) analyzing intensity, frequency, and trend variations of SSs; and (4) investigating compound impacts from SSs and other extreme events. Return levels of extreme SSs (Cid et al., 2018; Fang et al., 2021), sometimes storm

tides (Buchanan et al., 2017; Wahl et al., 2017; Dullaart et al., 2021) or total water levels (Kirezci et al., 2020; Boumis et al., 2023), estimated through the extreme value analysis theory (Coles, 2001) can be used to guide the design of coastal defense structures like seawalls and breakwater to reduce coastal flood risks in the future. This application heavily relies on the length of SS records; namely, the extrapolation of return periods cannot be longer than four times the length of available time series (Pugh and Woodworth, 2014). For example, at least 25-year SS data are needed to estimate 1 in 100-year SS levels since the

estimation uncertainty will increase if the records are too short. For the second category, coastal sea level variability is affected by multiple processes due to the coastline having complicated shapes and providing a boundary to ocean dynamics (Woodworth et al., 2019). Evaluating how various factors, including SSs, contribute to ESLs across different time scales and areas can help us better understand and prepare for this coastal hazard (Menéndez and Woodworth, 2010; Feng and Tsimplis, 2014; Melet et al., 2018; Almar et al., 2021; Lowe et al., 2021; Parker et al., 2023). This type of research does not have as

strict a requirement for the length of SS records as the former application, but the longer the duration and the higher the spatial coverage of SS data, the more useful information can be obtained. Changes in SSs also need to be discussed. Unlike the widely agreed impact of sea level rise on ESLs, the contribution from changes in SSs remains controversial. Therefore, current coastal planning practices still assumed SSs were stationary (Hinkel et al., 2014). Recently, several studies showed that SS frequency and magnitude trends were likely affected by internal climate variability, even over periods as long as 60 years (Calafat et al.,

2022; Tadesse et al., 2022; Feng et al., 2023). Further assessment of this phenomenon requires SS records long enough. In addition, spatial coverage of SS data must be guaranteed to analyze regional changes in SSs since the impacts of decadal signals vary from area to area. For the last category, potential risks caused by compound extreme events (for example, SS, extreme rainfall, heatwaves, and river discharge) in low-lying coastal areas are often much greater than by either in isolation, which should be given great attention (Wahl et al., 2015; Hermans et al., 2024; Zhou and Wang, 2024). In summary, high

spatial coverage and sufficiently long SS records are the basis for deepening our understanding of ESLs from multiple dimensions.

  There are three main ways to obtain high-frequency (at least hourly) SS information: tide gauge observation, numerical model simulation, and data-driven model reconstruction. Tide gauges (TGs) are the most reliable source of coastal water-level observations (Marcos et al., 2019). However, their distribution is sparse and uneven. For example, as the most complete high-

frequency TG collection currently, the Global Extreme Sea Level Analysis version 3 (GESLA-3) dataset included 5,119 stations around the world, most of which were distributed in North America, Europe, Japan, and Australia (Haigh et al., 2023). This always limits TG applications in the abovementioned fields, especially when analyzing global spatial variations of SSs or ESLs. In addition, though some of the oldest TG stations can date back to the eighteenth century, only ~10% of TG records



in the GESLA-3 dataset were longer than 50 years, which makes it relatively difficult to assess more detailed long-term variations in SSs globally. Numerical models can address the spatial coverage issue by resolving coastal shallow-water equations (Muis et al., 2016, 2023; Lockwood et al., 2024). However, numerical models are relatively time-consuming, especially for the global simulation. For instance, the state-of-the-art Global Tide and Surge Model (GTSM), whose simulation outputs have been widely used in relevant studies (Kirezci et al., 2020; Dullaart et al., 2021; Fang et al., 2021; Yang et al., 2024b), needed to take 21 days to complete 1-year simulation when 4 cores were used for a parallel setup (Muis et al., 2020a).

The total task required hundreds of thousands of core hours and tens of terabytes of storage (Muis et al., 2020b), which placed high demands on the performance of the computer cluster. The computational complexity of numerical models might affect the length of simulated SSs (Muis et al., 2019) and the data update period, which may impose some limitations on studies requiring long-term SS records. Data-driven models do not need to resolve physical coastal processes. They obtain the statistical relationship between SSs (predictand) and relevant atmospheric factors (predictor) through multiple linear regression

(Cid et al., 2018) or artificial intelligence (Bruneau et al., 2020), which can reduce computational complexity and hence effectively improving computational efficiency and the length of reconstructed SSs (Tadesse et al., 2020). For example, the Global Storm Surge Reconstruction (GSSR) database, the only publicly released global SS dataset from the data-driven model till now, provided SS reconstructions going as far back as 1836, which benefited the research of long-term trends in SSs (Tadesse and Wahl, 2021). However, because the model must establish independent relationships for every TG site by site

(Cid et al., 2017; Bruneau et al., 2020; Tiggeloven et al., 2021), it provided 882 points globally but cannot give any SS information at ungauged locations. To solve this issue, Yang et al. (2023) proposed a novel all-site modeling (ASM) framework for the data-driven model, which can reconstruct hourly SSs along the whole coastline by pooling all available TGs into one model. Evaluations on the regional scale showed that the precision of this framework was better than that of GSSR and was at the same level as that of the hydrodynamic model GTSM (Yang et al., 2023). Moreover, the training process can be completed

in several minutes (Yang et al., 2024b). Therefore, the ASM provides an opportunity to obtain global long-term and high spatial coverage SSs simply and efficiently.

The coastal areas within ~45°S to ~45°N are severely affected by SSs since most destructive tropical and extratropical cyclones occur here (Knapp et al., 2010). This research will use the ASM framework to establish a SS model in this area. After precision assessment by comparing it with TG observations and the numerical model GTSM, we will release, for the first time,

a long-term (> 80 years from 1940 to 2020) quasi-global hourly SS dataset reconstructed from the data-driven model with high spatial resolution (10 km along the coastline). Note that though the equatorial region (~6°S to ~6°N), the South Atlantic, and the southeastern Pacific have almost no tropical or extratropical cyclones, we also reconstruct SSs in these regions for data integrity. We hope this dataset, the ASM-SS (all-site modeling storm surge), will provide possible alternative support for coastal communities to deepen our understanding of SSs and ESLs.



## 2 Materials and Methods

### 2.1 Atmospheric Data

Atmospheric predictors from 1940 to 2020 were obtained from the European Centre for Medium-Range Weather Forecasts (ECMWF) Reanalysis v5 (ERA5) database (Hersbach et al., 2020). It is the fifth generation ECMWF reanalysis through assimilating model data with observations across the world into a globally complete and consistent dataset, which can provide hourly atmosphere fields with 0.25°×0.25° resolution grids. Following Yang et al.(2023) and Yang et al. (2024), four variables from ERA5 were used, including mean sea level pressure (mslp), 10 m northward and eastward wind (u10, v10), and 2 m temperature (t2m).

### 2.2 Tide Gauge Data

TG observations from 1940 to 2020 came from the high-frequency (15 minutes or one hour) GESLA-3 dataset collected from 36 international and national data providers (Haigh et al., 2023). This dataset unified the time units (to coordinated universal time) and length units (to meters) of water level records from different sources. In addition, the analysis flag was added to each TG record, making it convenient to select available sea-level data. However, a stricter quality control process is needed since some sites still contain datum jumps and outliers (Haigh et al., 2023). Detailed TG preprocessing is as follows:

(1) Coastal TG stations located between 45°S-45°N were selected (excluding the Mediterranean, Black, and Caspian Sea). Additionally, two stations at the southernmost tip of New Zealand were retained, though they are beyond 45°S;

(2) For the case that TG data was provided by different sources covering similar periods, the file with longer records was kept; for the case that the sea-level time series for the same site was split into different files, they were merged to obtain the longest possible records;

(3) TG data were resampled to hourly, and the analysis flag=1 (means 'use') was used to filter out the available data for each TG. Datum jumps were adjusted, and obvious outliers were removed through visual inspection. Then, 1315 stations with a length longer than one year remained (Figure 1);

(4) After removing the inter-annual mean sea-level variability from TG data through the annual moving average, the SS time series can be obtained by subtracting tides estimated from the Utide (Unified Tidal Analysis and Prediction Functions) package (Codiga, 2011), which can select the most important components from 146 tidal constituents through an automated decision tree;

(5) Finally, a 12-hour moving average was applied to SS data to limit possible remaining tidal signals due to small phase shifts in predicted tides (Tiggeloven et al., 2021; Yang et al., 2023).



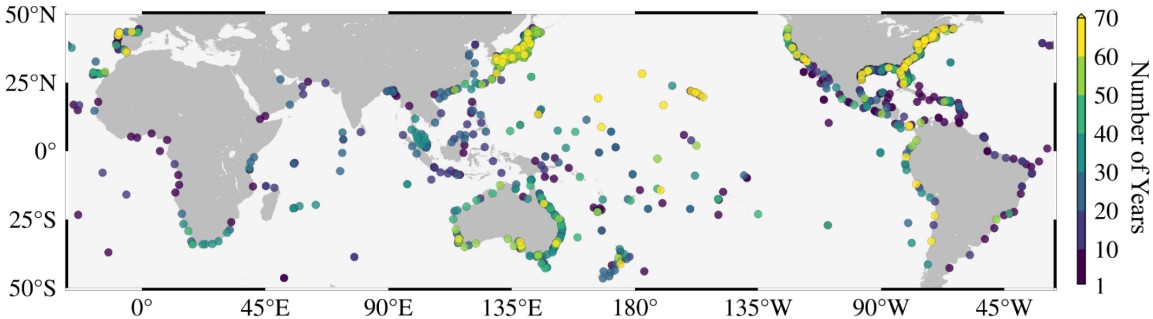

**Figure 1: The distribution and data length of selected tide gauges.**

## 2.3 Numerical Model Surge

Numerical model SSs came from GTSM version 3 global simulation forced with mean sea level pressure and wind from the ERA5 reanalysis (1979-2018), whose SS precision has been extensively evaluated and shown to have fair to good agreement with TG observations (Bloemendaal et al., 2019; Muis et al., 2020a; Parker et al., 2023; Yang et al., 2023). This model was solved based on Delft3D Flexible Mesh (Kernkamp et al., 2011) with the unstructured grid resolution from 2.5 km (1.25 km in Europe) along the coast to 25 km in the deep ocean (Muis et al., 2020a). It provided outputs both in the ocean and along the coastline; the latter's resolution was approximately every 20 km per coastal station.

## 2.4 Coastline Contour Data

The Global Self-consistent, Hierarchical, High-resolution Geography (GSHHG version 2.3.7) shoreline database (Wessel and Smith, 1996) was used to generate coastal stations for the ASM-SS in the research area (45°S to 45°N). The shoreline of this dataset was developed from the World Vector Shorelines and Atlas of the Cryosphere, providing five different-resolution coastline contours (crude, low, intermediate, high, and full). We used the high-resolution data (~300m). After smoothing the shoreline with a window of 50 points, coastal stations with a 10 km resolution were sampled evenly from the smoothed coastline. Figure 2 shows their distribution. The total number of stations is 20,440: Western Europe (200), Africa (2,806), North America (3,165), South America (2,218), Oceania (3,471), and Asia (8,580).

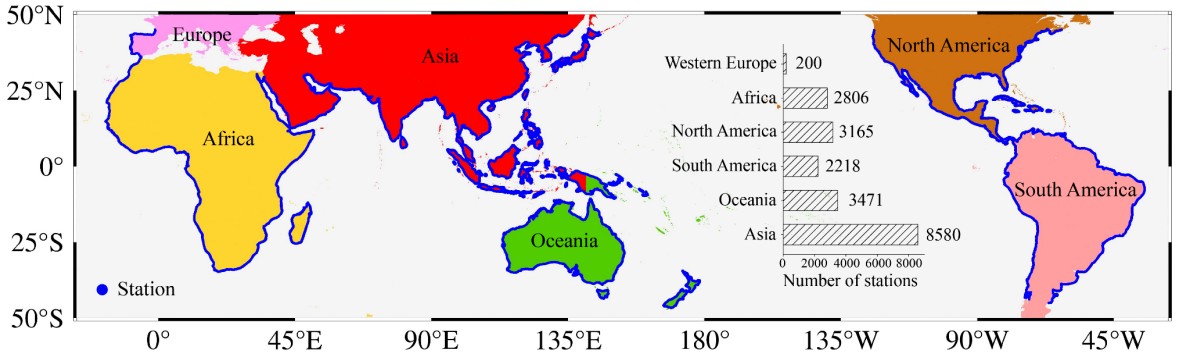

**Figure 2: The distribution of coastal stations for reconstructed storm surges.**



## 2.5 All-site Modeling Framework

Full details of the ASM can be found at Yang et al.(2023). Here, a conceptual description of it is provided. Assuming there are n available TGs within 45°S to 45°N, one data-driven model is established for all predictors from ERA5 and SSs from n TGs by the ASM. After learning the relationship through adequate training, this model can be used to reconstruct SSs at any gauged or ungauged coastal point within the research area by inputting relevant predictors. The following are the modeling processes:

(1) Obtaining predictors. Four atmospheric data (mslp, u10, v10, and t2m) for each TG station are extracted from the ERA5 dataset through linear interpolation. Meanwhile, following Yang et al.(2023) and Yang et al. (2024), TG locations and the time of their records are considered. Hence, the predictor matrix for each TG consists of 7 columns: mslp, u10, v10, t2m, longitude, latitude, and time;

(2) All-site modeling. Predictor matrices and SSs of all n TG stations are stacked into one predictor matrix and one SS matrix. Then, the eXtreme Gradient Boosting Tree (XGBoost) (Chen & Guestrin, 2016) is used to learn the relationship between these two matrices. The XGBoost is a residual machine learning model that generates a new decision tree using SS residuals from the previous tree. Therefore, the new tree will pay more attention to training where the residual errors are significant, making it suitable for modeling SS extremes;

(3) Reconstruction. SSs can be estimated for any coastline point by inputting the corresponding predictor matrix obtained following step (1) into the model established in step (2).

## 2.6 Model Performance Metrics

Three model performance metrics are used to evaluate the differences between reconstructed and observed SS levels: Pearson product-moment correlation coefficient (CORR), root mean square error (RMSE), and mean bias (MB):

$$CORR = \frac{\sum_{i=1}^{N}(SSL_{r,i} - \overline{SSL_r})(SSL_{o,i} - \overline{SSL_o})}{\sqrt{\sum_{i=1}^{N}(SSL_{r,i} - \overline{SSL_r})^2}\sqrt{\sum_{i=1}^{N}(SSL_{o,i} - \overline{SSL_o})^2}} \tag{1}$$

$$RMSE = \sqrt{\frac{\sum_{i=1}^{N}(SSL_{r,i} - SSL_{o,i})^2}{N}} \tag{2}$$

$$MB = \frac{1}{N}\sum_{i=1}^{N}(SSL_r - SSL_o) \tag{3}$$

where N is the length of the evaluation time series; $SSL_{r,i}$ and $SSL_{o,i}$ indicate the reconstructed and observed SS levels, respectively. $\overline{SSL_r}$ and $\overline{SSL_o}$ are the average values of them.

## 3 Results

### 3.1 Model Evaluation at Tide Gauges

The k-fold cross-validation strategy was chosen to evaluate the ASM model at TGs. 823 TG stations with time lengths exceeding 10 years between 1940 and 2020 were randomly divided into ten parts (i.e., 10-fold cross-validation), with the last



part containing 85 TGs. Each time, 9 of the parts were used for training. After the model was established, predictor matrices
of the excluded part of TGs were inputted into the model to obtain their SSs. The SSs of all parts of TGs can be estimated once
each part has been excluded. Then, we compared the reconstructed entire surge time series (evaluating the overall variation
trend) and the 95th percentile SSs (assessing extreme events) with TG observations. As shown in Figure 3, we divided the
research area by continent into six regions (WEU: Western Europe, AF: Africa, NA: North America, SA: South America, OC:

Oceania, and AS: Asia) for more detailed assessment information. In addition, as mentioned in the introduction, our analysis
here did not focus on the equatorial region (~6°S to ~6°N), the South Atlantic, and the southeastern Pacific. Evaluation results
for the entire area from 45°S to 45°N are presented in Fig A1 as an appendix.

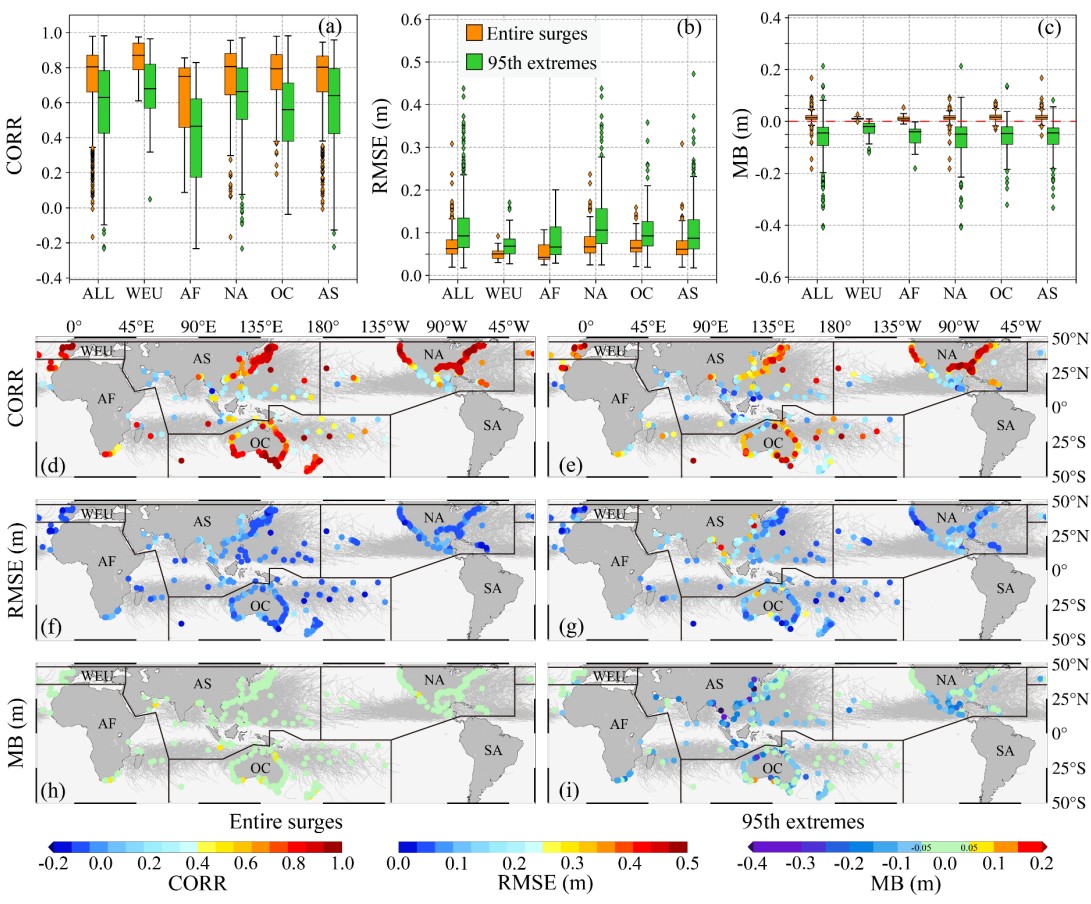

**Figure 3: Model evaluation at tide gauges from 1940 to 2020. (a-c) Entire surge and 95th extreme evaluation statistics for different**
180                                          **regions; (d-i) Distributions of evaluation metrics.**

Figure 3(a-c) show that for the entire time series of surges, whether on the quasi-global scale (i.e., for all TGs) or the regional
scale, medians of CORRs are around 0.8, RMSEs are better than 7 cm, and MBs are almost close to 0 cm. This means that the
overall variations of estimated surges from ASM confirm well with TG observations and that there is no significant systematic
bias (Figure 3(h)). In comparison, the reconstruction precision for extreme events (>95th percentile) is slightly lower: CORRs



are around 0.6 (with CORR<0.5 for AF), RMSEs are less than ~10 cm, and MBs are about -5 cm (indicating a slight underestimation of the magnitude of extreme events). Moreover, the precision for estimated extremes is better in regions with a relatively high density of TG stations, such as Western Europe, the United States, Japan, and Australia (Figure 3(e, g, i)). This result is consistent with the conclusion of Yang et al. (2024) that reducing the spatial interval of TG stations can benefit the estimation of extreme SSs.

**3.2 Comparison with Numerical Model at Tide Gauge Scale**

Considering that GTSM provided numerical surges from 1979 to 2018, the same period ASM data were extracted from SSs reconstructed in section 3.1. In addition, since points of GTSM did not completely coincide with TG stations, linear interpolation was used to interpolate GTSM SSs to the corresponding TG locations. Figure 4 gives the comparison results between ASM, GTSM, and TG observations (entire area results are in Fig A2). Note that we focused on extreme SSs in this

section as the overall variations of reconstructed SSs have been in good agreement with TGs.

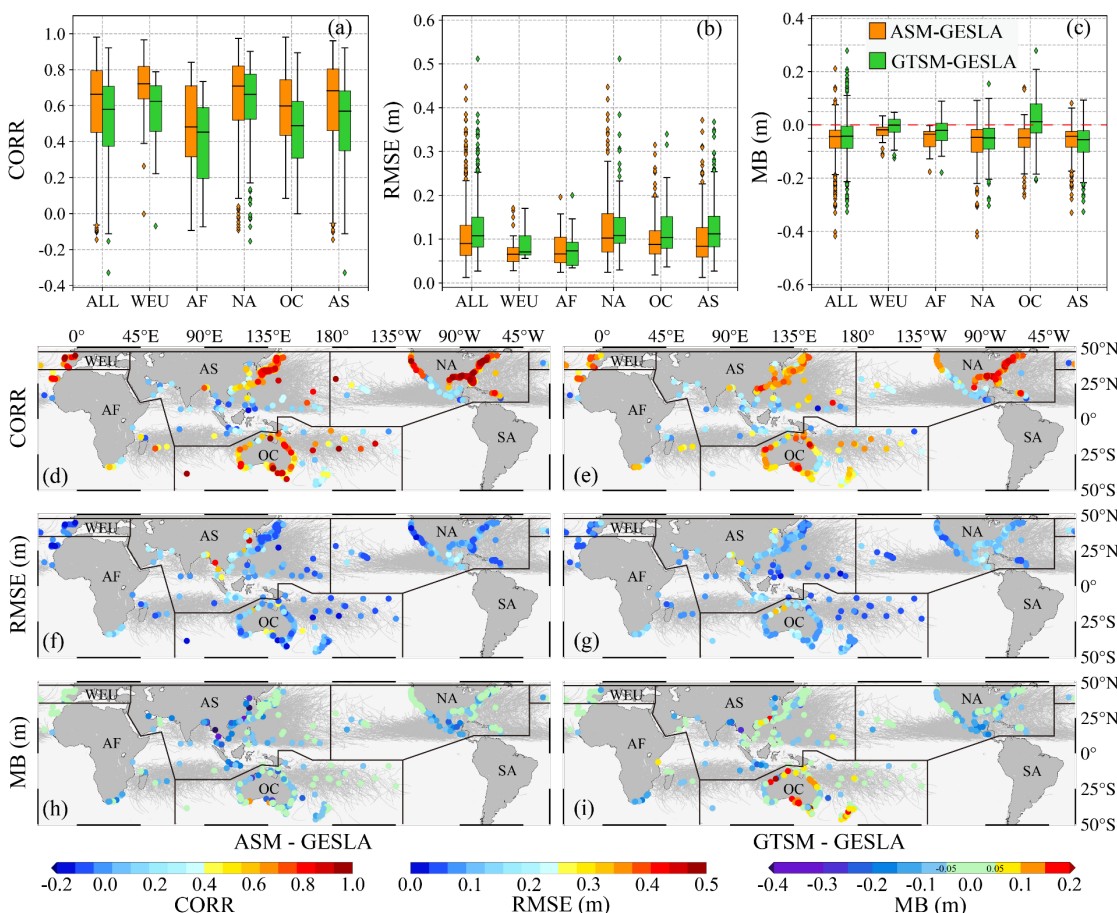

**Figure 4: Model comparison with the numerical model at tide gauges from 1979 to 2018. (a-c) ASM and GTSM 95th extreme evaluation statistics for different regions; (d-i) Distributions of evaluation metrics.**



It can be seen from Figure 4(a-c) that on the quasi-global scale, ASM (medians of CORRs, RMSEs, and MBs for 95th extremes

are 0.66, 9 cm, and -4.4 cm, respectively) slightly outperforms GTSM (medians are 0.58, 10.8 cm, and -4.3 cm). At the regional

scale, ASM and GTSM seem more precise in Western Europe, North America (especially the United States), East Asia, and

Oceania (Figure 4(d-i)). Moreover, CORRs and RMSEs of ASM are better than those of GTSM, while MBs of GTSM are

closer to 0 m in WEU, AF, and OC (Figure 4(a-c)). However, GTSM might appear to relatively overestimate extremes in some

regions, such as OC (Figure 4(i)). For further insight, Figure 5 presents scatter density plots of ASM and GTSM annual

maximum SSs compared with TG records in different regions. It is clearer that GTSM is slightly more likely to overestimate

extreme SSs (e.g., Figure 5(d, j, l)). In comparison, ASM is more stable than GTSM, with regional RMSEs reducing by around

5.3% (Figure 5(h) and Figure 5(g), RMSE of NA reducing from 0.187 m to 0.177 m) to 54.5% (Figure 5(d) and Figure 5(c),

RMSE of WEU decreasing from 0.187 m to 0.085 m). The overall RMSE and coefficient of determination ($R^2$) improvements

are 23.1% and 14.8%, respectively (Figure 5(b) and Figure 5(a)).

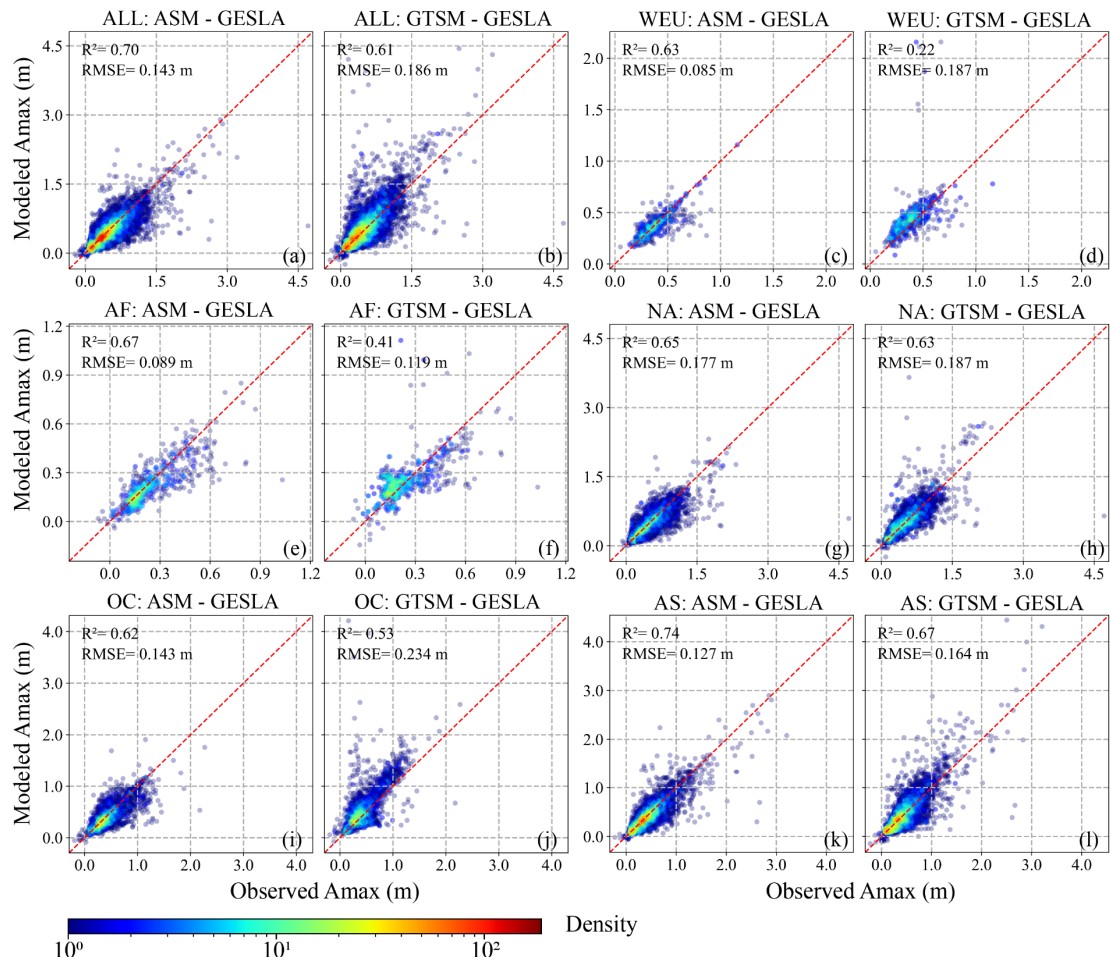

**Figure 5: Scatter density plots of ASM and GTSM annual maxima (Amax) compared with tide gauge observations in different regions. The data for tide gauges were combined. The red dotted line indicates the perfect fit line.**



## 3.3 Comparison with Numerical model at Coastal Scale

The comparison with TGs indicated that ASM is slightly better than GTSM in the research area. Moreover, the significant
advantages of the ASM data-driven model are: (1) compared to single-site data-driven models, it can provide SS information for ungauged points; (2) compared to numerical models, it can reconstruct long-term SSs faster. In this section, the ASM model was trained based on all 1,315 TGs with records longer than one year from 1940 to 2020 to generate SSs for ungauged locations. The training process consumed ~80GB of storage and took ~22 hours (~352 core hours) on a 16-core AMD EPYC 7317 processor, which showed a great advantage compared to the global numerical model GTSM (Muis et al., 2020b). After
reconstructing SSs (1979-2018) to all coastal points of GTSM to assess their differences, we generated the first long-term (>80 years from 1940 to 2020) quasi-global hourly data-driven SS dataset ASM-SS with 10 km spatial resolution along the coastline (see Figure 2).

Figure 6 (Fig A3 for the entire area) gives the comparison results between ASM and GTSM modeled entire surges and 95th extremes. Note that since both ASM and GTSM SSs were estimated, we used GTSM as the baseline here. As shown in Figure
6, there are relatively significant differences between ASM and GTSM. On the quasi-global scale, medians of CORRs, RMSEs, and MBs of the entire surges (95th extremes) between them are 0.39 (0.34), 8.6 cm (14.3 cm), and -5.4 cm (-13.1cm), respectively (Figure 6(a-c)). The negative MBs indicate that ASM tends to give slightly lower SS estimates than GTSM, which is consistent with the conclusion from the comparison with TGs in section 3.2. From the regional perspective, the agreement between ASM and GTSM (Figure 6(d, f, h) for entire surges, Figure 6(e, g, i) for 95th extremes) are relatively better in Western
Europe, North America (especially the United States), East Asia, and Oceania. For other places, on the one hand, both ASM and GTSM showed relatively poor agreement with TG observations in section 3.2 (Figure 4(d-i)); on the other hand, there are also visible discrepancies between ASM and GTSM (Figure 6(d,-i)). Possible reasons could be as follows: For ASM, its extreme SS reconstruction is affected by the distribution and spatial interval of TG stations (Yang et al., 2024b). For GTSM, the grid resolution and the precision of bathymetric data also have impacts on the simulation results. Additionally, neither of
them considers sea level variations caused by runoff and precipitation. Nevertheless, the precision of ASM and GTSM for these regions might need further improvement in the future.

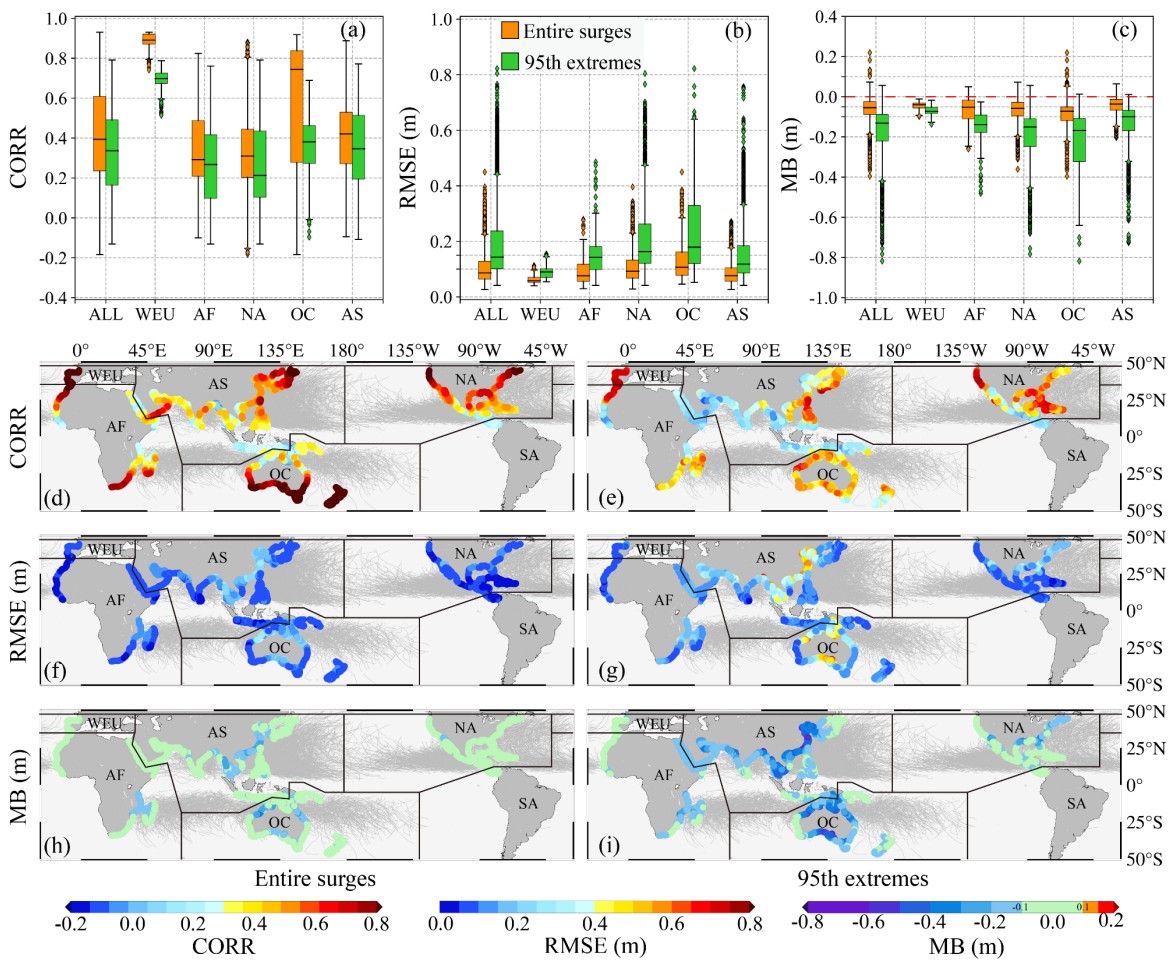

**Figure 6: Differences between ASM and GTSM at the coastal scale from 1979 to 2018. (a-c) Comparison statistics between ASM and GTSM modeled entire surges and 95th extremes for different regions; (d-i) Distributions of comparison metrics.**

## 4 Data availability

The ASM-SS quasi-global hourly data-driven storm surge dataset from 1940 to 2020 is available at https://doi.org/10.5281/zenodo.13293595 (Yang et al., 2024a) as NetCDF files month by month. The dataset includes five parameters: longitude, latitude, stations, time, and surge.

## 5 Conclusion and Discussion

High spatial coverage and long-term SS records are the basis for deepening our understanding and better preparing coastal communities for incoming ESLs. Due to the sparse and uneven distribution of TG stations, high spatial resolution SS information on a global or quasi-global scale could only be provided by global numerical models for a long time. Here, based





on the ASM framework, we established a SS model using observations from TGs between 45°S-45°N. Then, for the first time, a high spatial resolution (every 10 km per station along the coastline), long-term (over 80 years from 1940 to 2020), quasi-

global (within 45°S-45°N), hourly data-driven SS dataset ASM-SS was reconstructed from this model. Evaluation results indicate that for 95th extreme SSs, this model (medians of CORRs, RMSEs, and MBs are 0.66, 9 cm, and -4.4 cm, respectively) is slightly better than the state-of-the-art hydrodynamic model GTSM (medians are 0.58, 10.8 cm, and -4.3 cm); for annual maximum SSs, ASM is more stable than GTSM with overall RMSE and coefficient of determination optimizing by around 23.1% and 14.8%, respectively. This dataset could provide possible alternative support aside from numerical models for coastal

communities to evaluate return levels of extremes, assess the contribution of SSs to ESLs, analyze variations of SSs, investigate compound impacts from SSs with different extreme events, and other relevant applications.

Nonetheless, several details of this model can be studied more deeply in our future work: (1) Generally speaking, tropical and extratropical cyclones are usually accompanied by heavy rainfall when they make landfall, which might affect sea-surface height. In addition, the impact of river runoff in estuarine areas may need to be considered. (2) The distribution and spatial

interval of TG stations have been proven to affect the precision of ASM (Yang et al., 2024b). Because establishing and maintaining a permanent TG network with high spatial coverage in coastal regions is expensive and complex, it is necessary to consider integrating various water level observation technologies, such as Global Navigation Satellite System reflectometry (GNSS-R) and satellite altimetry. (3) From the predictor side, several studies showed that ERA5 data tends to relatively underestimate higher wind speeds (Graham et al., 2019; Xiong et al., 2022), which may lead to underestimations of extreme

SSs. Therefore, the atmospheric predictors can also be optimized through multi-source data fusion, such as considering wind speeds obtained from spaceborne GNSS-R (e.g., Cyclone Global Navigation Satellite System) or cyclone information obtained from remote sensing satellites.

# Appendix A



**Fig A1: Model evaluation at tide gauges from 1940 to 2020 for the whole area. (a-c) Entire surge and 95th extreme evaluation statistics for different regions; (d-i) Distributions of evaluation metrics.**



**Fig A2: Model comparison with the numerical model at tide gauges from 1979 to 2018 for the whole area. (a-c) ASM and GTSM 95th extreme evaluation statistics for different regions; (d-i) Distributions of evaluation metrics.**

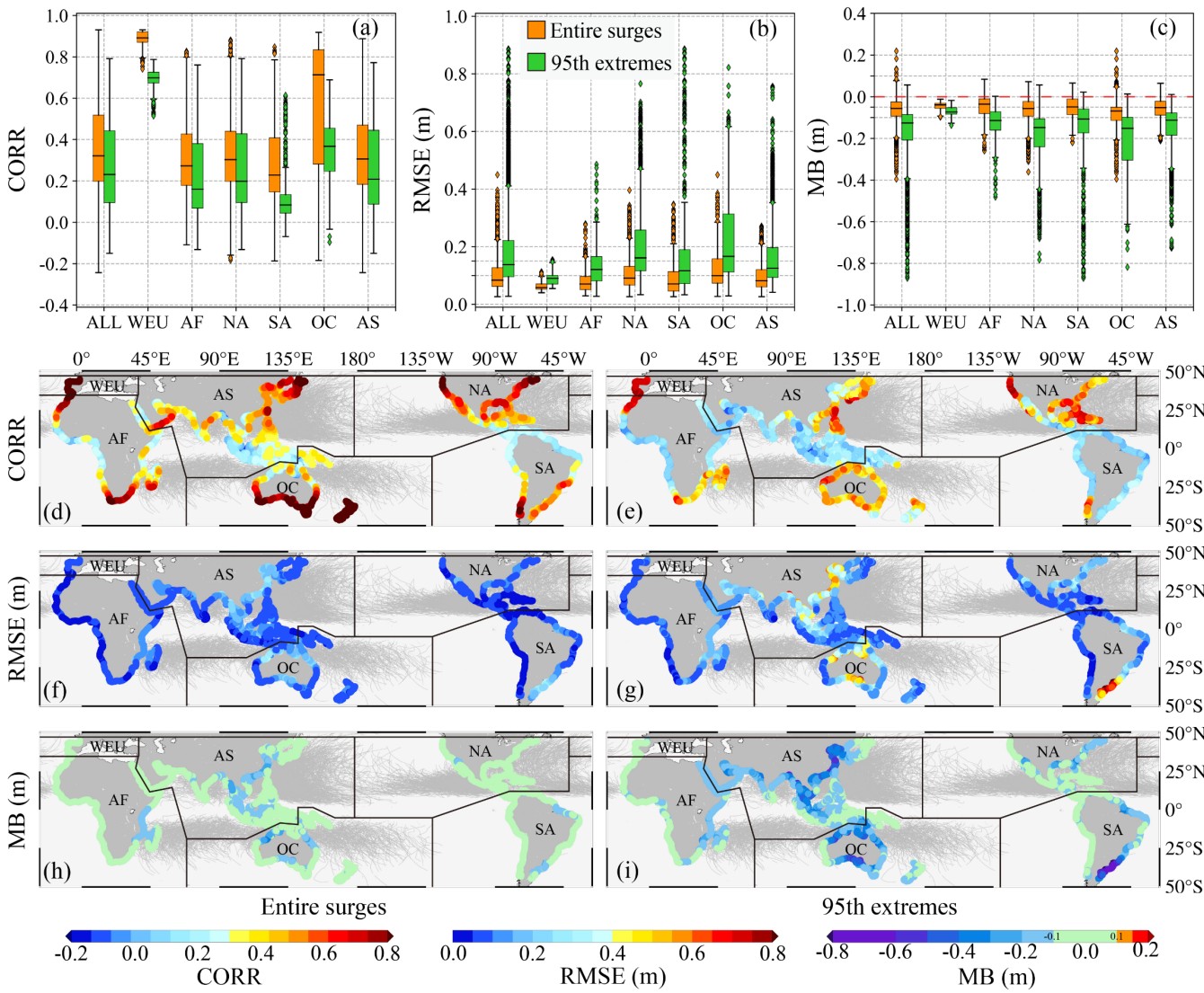


**Fig A3: Differences between ASM and GTSM at the coastal scale from 1979 to 2018 for the whole area. (a-c) Comparison statistics between ASM and GTSM modeled entire surges and 95th extremes for different regions; (d-i) Distributions of comparison metrics**

**Author contribution.** LY and TJ designed the research. LY carried out the experimental results and wrote the initial manuscript. TJ and WJ provided related comments for this work and revised the manuscript.

**Competing interests.** The authors declare that they have no conflict of interest.



**Acknowledgments.** The authors are very grateful to the Climate Data Store for providing ERA5 (Copernicus Climate Change Service, 2018) and GTSM data (Copernicus Climate Change Service, 2022). We would also like to thank the publication of the GESLA-3 dataset, which helps us save much time in tide gauge collection and data preprocessing. (https://gesla787883612.wordpress.com, last access: 12 August 2024). The shoreline database GSSHS version 2.3.7 is available online (https://www.ngdc.noaa.gov/mgg/shorelines, last access: 12 August 2024). All the respectable reviewers and editors are acknowledged for their professional suggestions for this paper.

**Financial support.** This research was funded by the National Natural Science Foundation of China under Grants 42374035 and 42192531, and the Fundamental Research Funds for the Central Universities.

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
