# Peer review of "ASM-SS: The First Quasi-Global High Spatial Resolution Coastal Storm Surge Dataset Reconstructed from Tide Gauge Records"

_Earth System Science Data, 2024_

## Author Comment (AC1)

**Response to Reviewer#1 Comments**

**Recommendation** – a major revision is required before the paper can be published.

The paper describes a new dataset of storm-surges. The methodology is well described and written. However, a main issue needs to be addressed before the paper can be recommended for publication. The purpose of the dataset development is not well grounded and presented.

**Point 1:** For example, section 3.3 opens with:

"The comparison with TGs indicated that ASM is slightly better than GTSM in the research area. Moreover, the significant advantages of the ASM data-driven model are: (1) compared to single-site data-driven models, it can provide SS information for ungauged points; (2) compared to numerical models, it can reconstruct long-term SSs faster."

These are the advantages of the model while the paper should focus on the benefits of the dataset. One can argue that a single-site data model can provide higher accuracy, and numerical models can also give results in ungauged points and with higher resolution than the newly developed dataset. In which case, who are the target users for the dataset?

The paper should be revised to clarify the advantages and novelty of the dataset.

**Response:**

Thanks for your constructive suggestions. High spatiotemporal resolution and sufficiently long SS dataset is the basis for analyzing this disaster. However, the existing storm surge datasets cannot fulfill all three demands simultaneously on a global or quasi-global scale. The advantage and novelty of this dataset is that it can fix the gaps among tide gauge observations, global numerical model simulations, and data-driven reconstructions from single-site modeling. We meticulously revised this manuscript based on the reviewer's suggestions, hoping this version can be clearer and more logical:

**Point 2:** Here are some points to consider: One of the key features is the spanning. Does the accuracy change over time? It must vary as it is directly affected by the ERA5 inputs.

**Response:**

Thanks for your professional suggestion. In this version, we added the temporal variation analysis of our model's precision from 1940 to 2020 every 10 years in Lines

It is necessary to evaluate temporal variations in reconstructed SSs further since their length is over 80 years, during which the number of TG stations and the quality of atmospheric data have changed. As shown in Fig. 4, the precision of ASM model at TGs in each sub-region was calculated every 10 years (excluding TGs with less than one year of data in a given decade). Results indicate that the overall precision (i.e., for ALL TGs) of entire surges and 95th extremes gradually increased from 1940 to 2020. Possible reasons are, on the one hand, the increase of TGs in recent decades provided more predictand features; on the other hand, the optimization of ERA5 atmospheric data (predictor) contained more detailed tropical and extratropical cyclone information. At the regional scale, for entire surges, Fig. 4(a) indicates that except for SWA (CORR decreases) and WAS (CORR remains unchanged), CORRs of other sub-regions present an upward trend; Fig. 4(b) shows the RMSE in SES increases, while RMSEs in other regions decrease; Fig. 4(c) gives that MBs of sub-regions have been gradually optimized (excluding WAS). For 95th extremes, in terms of CORR (Fig. 4(d)), WEU, NAF, WNA, ENA, EAS, NOC, and SOC show an upward trend, whereas there is no obvious pattern in other regions; for RMSE (Fig. 4(e)), ER, SEA, and SES present an increasing trend, other regions decrease; for MB (Fig. 4(f)), the underestimation of SSs in ER and SAS rises, and there is no noticeable change in WNA and SES. MBs in WEU, NAF, ENA, WAS, EAS, NOC, and SOC are optimized, while there is no clear pattern in SWA, SEA, CA, and SWS.

[Figure]

Figure 4: Temporal variations of the ASM model's precision at tide gauges from 1940 to 2020. (a-c) Entire surge evaluation statistics for different regions every 10 years; (d-f) 95th extreme evaluation statistics for different regions every 10 years

**Point 3:** The discussion of spatial variability in accuracy needs elaboration. The boxed continental statistics look nice but do not address all the shown differences. For example, performance in U.S. and Canada coasts are much better than in Central America. That could be attributed to the climatology, forcing fields or TG availability.

**Response:**

Thanks for your recommendation. In this version, we divided the research area into more detailed sub-regions (from six in the old version to fifteen in this version), and all relevant figures and descriptions were updated:

As shown in Fig. 3, we divided the research area into fifteen sub-regions (ER: the equatorial region, WEU: Western Europe, NAF: Northern Africa, SWA: Southwestern Africa, SEA: Southeastern Africa, WNA: Western North America, ENA: Eastern North

America, CA: Central America, SWS: Southwestern South America, SES: Southeastern South America, WAS: Western Asia, EAS: Eastern Asia, SAS: Southern Asia, NOC: Northern Oceania, and SOC: Southern Oceania) for more detailed assessment information. Note that the equatorial region (~6°S to ~6°N) was separated as an independent area since it has almost no tropical or extratropical cyclones.

[Figure]

Figure 3: ASM model evaluation at tide gauges from 1940 to 2020. (a-c) Entire surge and 95th extreme evaluation statistics for different regions; (d-i) Distributions of evaluation metrics. Gray lines are tropical cyclone paths.

[Figure]

Figure 5: ASM model comparison with the numerical model at tide gauges from 1979 to 2018. (a-c) ASM and GTSM 95th extreme evaluation statistics for different regions; (d-i) Distributions of evaluation metrics. Gray lines are tropical cyclone paths.

[Figure]

Figure 6: Scatter density plots of ASM and GTSM annual maxima (Amax) compared with tide gauge observations in different regions. The data for tide gauges were combined. The red dotted line indicates the perfect fit line.

[Figure]

Figure 7: Differences between ASM and GTSM at the coastal scale from 1979 to 2018. (a-c) Comparison statistics between ASM and GTSM modeled entire surges and 95th extremes for different regions; (d-i) Distributions of comparison metrics. Gray lines are tropical cyclone paths.

**Point 4:** Adding for each output point a measure of accuracy/quality based on the author's evaluations will be of much help to the users.

**Response:**

Thanks for your suggestion. The evaluation of accuracy relies on the ground truth from tide gauge stations. However, the spatial resolution of our dataset is 10 km, which means there are no tide gauge stations nearby for most of the output points. The cross-validation in section 3.1 can offer such an assessment. We divided the research area into more detailed parts (see Point 3) and hope it can provide users with more helpful quality

information.

**Point 5:** Excluding most of Europe devalues the significance of the dataset as it is supposed to be "quasi-global".

**Response:**

Thanks for reminding. Since this was the first high spatial coverage dataset generated through the data-driven method over such a large area, we wanted to focus on low-latitude regions where tropical cyclones are frequent. In addition, the Mediterranean Sea was excluded because we found almost no tropical cyclones in this area's history. These two conditions excluded most of Europe from the ASM-SS dataset. In future dataset updates, we will try to expand the data coverage.

**Specific Remarks:**

**Point 6:** Lines 46-47: "Changes in SSs also need to be discussed. Unlike the widely agreed impact of sea level rise on ESLs, the contribution from changes in SSs remains controversial." The sentence is unclear and confusing. Please, rewrite it.

**Response:**

We are sorry for the unclear expression. This version deleted this sentence since we rewrote nearly the whole introduction.

**Point 7:** Line 57: "There are three main ways to obtain high-frequency...". Listing the three ways together is problematic as data-driven models are dependent on TG observations. Additionally, authors' data-driven model depends also on a numerical model. It would be great to introduce the concept of the data-driven model afterwards.

**Response:**

Thanks for your professional suggestion. We rewrote the relevant paragraphs in Lines 31-74, hoping it could be more logical:

[revised manuscript text omitted]

**Point 8:** Line 66: "However, numerical models are relatively time-consuming, especially for the global simulation." Time-consuming is not the most concerning matter for the dataset users. What are the disadvantages of numerical models related to their performance?

**Response:**

Thanks for your professional recommendation. Precise numerical simulations require accurate and high-resolution bathymetric data, which is often unavailable in nearshore areas. In addition, the coastal grid resolution for global or quasi-global numerical models is usually set to several kilometers, which means small-scale physical features cannot be sufficiently simulated, and hence affecting the precision. We rewrote this paragraph in Lines 41-52:

Numerical models can provide simulated data with better spatial coverage by resolving coastal physical processes inducing SSs (Muis et al., 2016, 2023; Lockwood

et al., 2024). A common limitation of numerical models is that they require accurate and high-resolution bathymetric data for sufficiently precise SS estimations since SSs are significantly affected by water depth in shallow water (Resio and Westerink, 2008). However, such bathymetric data is often unavailable in nearshore areas (Cid et al., 2018). In addition, in global or quasi-global SS simulations, the coastal grid resolution of numerical models is usually set to several kilometers to balance the computational complexity (Muis et al., 2020; Mentaschi et al., 2023), which means that nearshore physical features with a spatial scale smaller than this resolution cannot be sufficiently simulated (Parker et al., 2023), and hence affecting the SS precision. Meanwhile, the computational efficiency of global numerical models tends to affect the length of simulated SSs (Muis et al., 2019). For instance, the state-of-the-art Global Tide and Surge Model (GTSM), though its outputs have been widely used in relevant studies (Kirezci et al., 2020; Dullaart et al., 2021; Fang et al., 2021; Yang et al., 2024b), its simulations spaned only the most recent decades from 1979 to 2018 (Muis et al., 2020). This imposed limitations on studies requiring long-term SS records.

**Point 9:** Line 85: "Therefore, the ASM provides an opportunity to obtain global long-term and high spatial coverage SSs simply and efficiently." This para emphasizes the effectiveness of the model rather than ensuring that data-driven models provide reliable results. Please, rewrite it.

**Response:**

Thanks for your suggestion. We rewrote this paragraph in Lines 53-74:

Unlike numerical models, data-driven models do not need to resolve coastal physical processes. They obtain the statistical relationship between SSs (predictand) and relevant atmospheric factors (predictor) through multiple linear regression (Cid et al., 2018) or artificial intelligence (Bruneau et al., 2020). Therefore, the precision of data-driven models is unaffected by bathymetric data and grid resolution. In addition, long-term SSs can be reconstructed efficiently after the statistical relationship is established (Tadesse et al., 2020). However, the commonly used single-site modeling framework for data-driven models heavily relies on TGs; it must establish independent relationships for every TG site by site (Cid et al., 2017; Bruneau et al., 2020; Tiggeloven et al., 2021) and cannot provide any SS information at ungauged coastal locations. For example, the Global Storm Surge Reconstruction (GSSR) database, the only publicly released global SS dataset from the data-driven model, provided SS reconstructions at 882 points globally going as far back as 1836, which benefited the research on longterm trend analysis of SSs (Tadesse and Wahl, 2021). However, it cannot address issues caused by the sparseness and uneven distribution of TG stations. Some studies replaced TG observations with numerical SS simulations to train the data-driven model (so-called 'surrogate model') (Lee et al., 2021; Ayyad et al., 2022; Lockwood et al., 2022). This combination improved the spatial resolution, but numerical models' precision limitations were also transferred to the surrogate model. Moreover, in theory, surrogate models cannot be better than numerical models compared to TG observations. Yang et al. (2023) proposed a novel all-site modeling (ASM) framework, which allowed the data-driven model to reconstruct high spatial-coverage SSs in research areas by learning from TG observations (without SS simulations from numerical models). Although single-site modeling and ASM belong to the data-driven model, their basic ideas differ. The former presumes SS observations at different TGs are independent. Therefore, the relationship between predictors and SSs needs to be learned for every TG site by site; this relationship is unsuitable for other locations. In contrast, the latter believes there is a universal connection between SSs at different TGs, so all available TGs within the research area can be pooled into one model to learn the only regional relationship between predictors and SSs. This essential difference enables the ASM framework to reconstruct SSs at any coastal point in the research area. In addition, the study has shown that ASM's precision is better than single-site modeling's (Yang et al., 2023).

**Point 10:** Line 95: The section opens with describing the inputs to the model. It would be better to start by explaining the methodology in general.

**Response:**

Thanks for your suggestion. We chose this logic since we can give the specific composition of each column of matrices when explaining the methodology, which may make it easier for readers to understand the modeling process. Therefore, we retained this logic in this version.

**Point 11:** Line 98: https://doi.org/10.1002/qj.4803 should be also referenced since you use the extended version of ERA5.

**Response:**

Thanks for your attention. We are sorry for forgetting to add it when submitting the manuscript since the newest paper of ERA5 was not published when we started to write our manuscript. It was updated in this version in Line 87:

…from the European Centre for Medium-Range Weather Forecasts (ECMWF)

Reanalysis v5 (ERA5) database (Soci et al., 2024)…

**Point 12:** Line 131: "every 20 km per coastal station." The choice of the 10km resolution in the developed dataset is unclear and not explained. The Numerical model resolution is 2.5 and the data-driven model resolution is 20km yet the dataset is provided at 10km. Please justify this choice.

**Response:**

We are sorry for the confusion. Though the numerical model was solved with the unstructured grid resolution of 2.5 km, the spatial resolution along the coast was resampled to ~20 km when the numerical product was released (considering the data volume). Hence, when we compared the data-driven model with the numerical model at the coastal scale, we needed to reconstruct storm surges to the same coastal points of the numerical product, whose spatial resolution is 20 km. Then, in order to distinguish our data-driven dataset from the global numerical product, we generated our storm surge product with a resolution of 10 km. A higher spatial resolution may provide more information for data users.

To make it clearer, we rewrote this sentence in Lines 121-122:

…It provided outputs both in the ocean and along the coastline; the latter's resolution was resampled to approximately every 20 km per coastal station to limit the data volume (Muis et al., 2020).

**Point 13:** Line 137: "coastal stations with a 10 km resolution" the using term station here is confusing. Please, find a different way to call the model results (maybe nodes).

**Response:**

We are sorry for the confusion. We changed it in this version in Line 129-133:

…After smoothing the shoreline with a window of 50 points, coastal nodes with a 10 km resolution were sampled evenly from the smoothed coastline. Figure 2 shows their distribution. The total number of nodes is…

[Figure]

Figure 2: The distribution of coastal nodes for reconstructed storm surges.

**Point 14:** Line 175: "In addition, as mentioned in the introduction, our analysis here did not focus on the equatorial region (~6°S to ~6°N), the South Atlantic, and the southeastern Pacific." If these areas are a part the published dataset they must be evaluated as well. It is suggested to have the evaluations of the entire domain as the primary and the limited evaluations in the appendix.

**Response:**

Thanks for your constructive suggestions. In this version, we moved the evaluations of the entire domain from the appendix to the main text and divided the research area into more specific parts. We rewrote this paragraph in Lines 173-181:

As shown in Fig. 3, we divided the research area into fifteen sub-regions (ER: the equatorial region, WEU: Western Europe, NAF: Northern Africa, SWA: Southwestern Africa, SEA: Southeastern Africa, WNA: Western North America, ENA: Eastern North America, CA: Central America, SWS: Southwestern South America, SES: Southeastern South America, WAS: Western Asia, EAS: Eastern Asia, SAS: Southern Asia, NOC: Northern Oceania, and SOC: Southern Oceania) for more detailed assessment information. Note that the equatorial region (~6°S to ~6°N) was separated as an independent area since it has almost no tropical or extratropical cyclones.

[Figure]

Figure 3: ASM model evaluation at tide gauges from 1940 to 2020. (a-c) Entire surge and 95th extreme evaluation statistics for different regions; (d-i) Distributions of evaluation metrics. Gray lines are tropical cyclone paths.

**Point 15:** Line 180: Fig 3 caption says "Model evaluation at tide gauges..". Since there are several models involved in the work there should be a naming consistency to avoid confusion.

**Response:**

We are sorry for the inaccurate wording. We rewrote it in Line 180 and Line 213:

Figure 3: ASM model evaluation at tide gauges from 1940 to 2020….

Figure 4: ASM model comparison with the numerical model at tide gauges…

**Point 16:** Line 214: "the significant advantages of the ASM data-driven model are: (1)

compared to single-site data-driven models, it can provide SS information for ungauged points; (2) compared to numerical models, it can reconstruct long-term SSs faster. In this section" As it was mentioned above, these advantages are first and foremost related to the model itself rather than the developed dataset and its preference compare to the similar datasets.

**Response:**

Thanks for your constructive suggestions. In this version, we focused on clarifying the dataset's advantages. This description was changed in Lines 239-243 into:

As mentioned in the introduction, though ASM and single-site modeling belong to the data-driven model, the former can provide SS information for ungauged points since their basic ideas differ. This advantage of ASM allows us to compare the data-driven model and numerical model on a quasi-global scale with high spatial resolution. In this section, the ASM model was trained based on all 1,315 TGs within the research area with records longer than one year from 1940 to 2020 (Fig. 1). Then SSs from 1979 to 2018 were reconstructed to all coastal points of GTSM to assess their differences (Fig. 7).

**Point 17:** Appendix A: Either add text to the appendix or move the figures to the sections which mention them.

**Response:**

Thanks for your suggestion. In this version, we deleted Appendix A and analyzed the whole area in relevant sections.

Overall, the authors showed the new dataset has potential. But the paper requires significant improvement to be published.

---

## Author Comment (AC2)

**Response to Reviewer#2 Comments**

The manuscript uses machine learning methods to establish relationships between tide gauge measurements and several atmospheric and oceanic variables, generating a global coastal storm surge dataset at 10 km spatial resolution. Overall, the generated dataset is of substantial application value, and the validation results show strong performance, particularly in the reconstruction of extreme values—a known challenge for AI models. The topic aligns well with the aims of ESSD. However, there are several key areas that require attention to ensure the manuscript is clear, methodologically sound, and accessible to readers.

**Majors:**

**Point 1:** The discussion of previous studies in the introduction lacks depth. The authors list previous studies without effectively explaining how the current work advances the field. To strengthen this section, the introduction should focus more on the existing gaps in storm surge modeling and how the proposed dataset addresses those shortcomings. The classification of storm surge research is overly simplified. The four categories mentioned in the second paragraph overlap and include one another. Moreover, the machine learning approach presented in this paper is described as separate from AI-based methods, though it clearly falls within that domain as a regression model. The difference between this approach and single-site models is primarily in the inputs used, such as geographic and temporal variables, but the fundamental methodology remains similar. A more refined categorization would provide better context for the reader.

**Response:**

Thanks for your constructive suggestions. In this version, we deleted the second paragraph and rewrote the introduction in Lines 31-84 to make it focus on the existing gaps in storm surge datasets and how our method can fix the gaps. The logic is from tide gauge observations to numerical models, then to data-driven models:

[revised manuscript text omitted]

**Point 2:** The description of the model's methodology lacks sufficient detail on its innovations. For instance, the choice of specific atmospheric and oceanic variables from ERA5 should be justified, and the process of integrating geographical and temporal variables requires further explanation. How were these inputs pre-processed to allow for prediction across any coastal location or time? This is a key aspect of the model and should be clarified. Although more detailed explanations may have been presented in the authors' previous publications, it is still important to concisely convey these methodological details in this data-focused paper to ensure readers can fully understand the process without referring to other sources.

**Response:**

We are sorry for any difficulties in understanding. We added more details of our modeling framework in this version (Lines 135-150), hoping it can be clearer and more understandable to readers:

Full details of the ASM can be found at Yang et al.(2023). Here, a conceptual description of it is provided. As mentioned in the introduction, though single-site modeling and ASM belong to the data-driven model, their basic ideas are different. For example, assuming there are n available TGs within 45°S to 45°N. The single-site modeling presumes SS observations at n TGs are independent; each site needs to build a separate data-driven model to learn the relationship between predictors and SSs at that station. In this case, n single-site modeling data-driven models are established, and they cannot reconstruct SSs for locations other than TG stations. Unlike single-site modeling, the ASM believes a general connection exists between SSs at n TGs within the research area. Namely, there is a unique regional relationship between predictors and SSs, and all TGs follow this relationship. Therefore, predictors and SSs at n available TGs can be pooled into one ASM data-driven model. After learning the only regional relationship through adequate training, this ASM model can be used to reconstruct SSs at any gauged or ungauged coastal point within the research area by inputting relevant predictors. The following are the modeling processes:

(1) Obtaining predictors. Four atmospheric data (mslp, u10, v10, and t2m) for each TG station are extracted from the ERA5 dataset through linear interpolation. Changes in sea level pressure and wind are the main factors in generating SSs (Woodworth et al., 2019); adding temperature variations considers the effects of thermal expansion and contraction. Meanwhile, following Yang et al.(2023) and Yang et al. (2024a), another three variables (longitude, latitude, and timestamp) are considered since geographical locations and record lengths of TGs are different. Hence, the predictor matrix for each

TG consists of 7 columns: mslp, u10, v10, t2m, longitude, latitude, and time;

**Point 3:** One of the key strengths of the model is its superior performance in predicting extreme storm surge events compared to numerical models. However, the reasons behind this superior performance are not fully explored. A deeper analysis of why the machine learning model performs better than numerical models in extreme cases, particularly considering that AI models often struggle with extremes, would add significant value.

**Response:**

Thanks for your suggestion. In this version, we discussed the possible reason in Lines 228-234 after the comparison between our data-driven mode and numerical model in section 3.2:

The reason why ASM outperforms GTSM can be attributed to two main aspects. For the global numerical model GTSM, as mentioned in the introduction, the accuracy and spatial resolution of bathymetric data in the nearshore area limits the precision of SSs. Meanwhile, the grid with a resolution of several kilometers affects the effective simulation of small-scale physical factors. For the ASM data-driven model, the training process is based on TG observations. TGs are the most accurate source for sea level monitoring, and their records can be considered to include effects from all spatial-scale physical processes. In addition, the machine learning method XGBoost is a residual model that pays more attention to where residual errors are significant, which also benefits the estimation of extreme SSs.

**Point 4:** While the manuscript provides a thorough discussion of the spatial performance of the dataset, it lacks an analysis of the model's temporal performance. How does the model perform over the 1940–2020 period? Are there periods when the model is more or less accurate? Providing this temporal analysis would add an important dimension to the validation results.

**Response:**

Thanks for your professional suggestion. In this version, we added the temporal variation analysis of our model's precision from 1940 to 2020 every 10 years in Lines 190-206:

It is necessary to evaluate temporal variations in reconstructed SSs further since their length is over 80 years, during which the number of TG stations and the quality of atmospheric data have changed. As shown in Fig. 4, the precision of ASM model at

TGs in each sub-region was calculated every 10 years (excluding TGs with less than one year of data in a given decade). Results indicate that the overall precision (i.e., for ALL TGs) of entire surges and 95th extremes gradually increased from 1940 to 2020. Possible reasons are, on the one hand, the increase of TGs in recent decades provided more predictand features; on the other hand, the optimization of ERA5 atmospheric data (predictor) contained more detailed tropical and extratropical cyclone information. At the regional scale, for entire surges, Fig. 4(a) indicates that except for SWA (CORR decreases) and WAS (CORR remains unchanged), CORRs of other sub-regions present an upward trend; Fig. 4(b) shows the RMSE in SES increases, while RMSEs in other regions decrease; Fig. 4(c) gives that MBs of sub-regions have been gradually optimized (excluding WAS). For 95th extremes, in terms of CORR (Fig. 4(d)), WEU, NAF, WNA, ENA, EAS, NOC, and SOC show an upward trend, whereas there is no obvious pattern in other regions; for RMSE (Fig. 4(e)), ER, SEA, and SES present an increasing trend, other regions decrease; for MB (Fig. 4(f)), the underestimation of SSs in ER and SAS rises, and there is no noticeable change in WNA and SES. MBs in WEU, NAF, ENA, WAS, EAS, NOC, and SOC are optimized, while there is no clear pattern in SWA, SEA, CA, and SWS.

[Figure]

Figure 4: Temporal variations of the ASM model's precision at tide gauges from 1940 to 2020. (a-c) Entire surge evaluation statistics for different regions every 10 years; (d-f) 95th extreme evaluation statistics for different regions every 10 years

**Point 5:** Figure 1 shows several tide gauge stations in South America and West Africa with long records, yet these regions are not featured in the validation results. The authors should explain why results from these areas were excluded from the analysis.

**Response:**

Apologies for this confusion. Our initial logic was to analyze the areas mainly affected by tropical or extratropical cyclones in the main text (since there are almost no tropical or extratropical cyclones in the equatorial region, the South Atlantic and the southeastern Pacific), and put the evaluation results of the entire area in the appendix. However, this did not seem to be a suitable way. Therefore, in this version, we deleted the appendix and moved the assessment of the entire domain into the main text for discussion. All relevant figures were updated:

As shown in Fig. 3, we divided the research area into fifteen sub-regions (ER: the

equatorial region, WEU: Western Europe, NAF: Northern Africa, SWA: Southwestern Africa, SEA: Southeastern Africa, WNA: Western North America, ENA: Eastern North America, CA: Central America, SWS: Southwestern South America, SES: Southeastern South America, WAS: Western Asia, EAS: Eastern Asia, SAS: Southern Asia, NOC: Northern Oceania, and SOC: Southern Oceania) for more detailed assessment information. Note that the equatorial region (~6°S to ~6°N) was separated as an independent area since it has almost no tropical or extratropical cyclones.

[Figure]

Figure 3: ASM model evaluation at tide gauges from 1940 to 2020. (a-c) Entire surge and 95th extreme evaluation statistics for different regions; (d-i) Distributions of evaluation metrics. Gray lines are tropical cyclone paths.

[Figure]

Figure 5: ASM model comparison with the numerical model at tide gauges from 1979 to 2018. (a-c) ASM and GTSM 95th extreme evaluation statistics for different regions; (d-i) Distributions of evaluation metrics. Gray lines are tropical cyclone paths.

[Figure]

Figure 6: Scatter density plots of ASM and GTSM annual maxima (Amax) compared with tide gauge observations in different regions. The data for tide gauges were combined. The red dotted line indicates the perfect fit line.

[Figure]

Figure 7: Differences between ASM and GTSM at the coastal scale from 1979 to 2018. (a-c) Comparison statistics between ASM and GTSM modeled entire surges and 95th extremes for different regions; (d-i) Distributions of comparison metrics. Gray lines are tropical cyclone paths.

**Point 6:** The manuscript suffers from imprecise language and grammatical errors. Phrases like "coastline having complicated shapes" (line 41) and "internal climate variability" (line 49) are vague and not commonly used in geoscience literature. Additionally, phrases such as "numerical models are based on shallow water equations" (line 65) overly simplify the complexity of these models. Grammatical issues such as "until now" (line 9) and "will" (line 89) create ambiguity and should be corrected for clarity. Moreover, the manuscript contains an excessive number of speculative terms such as "some," "might," "may," and "slightly better." Scientific writing should avoid

this level of uncertainty when possible, and more precise language should be used. Where quantifiable data are available, the authors should provide specific numbers to reduce ambiguity.

**Response:**

We are sorry for our imprecise language and grammatical errors. Phrases "coastline having complicated shapes" and "internal climate variability" were deleted since we rewrote the introduction; "numerical models are based on shallow water equations" was replaced with "…by resolving coastal physical processes inducing SSs" in Line 41. In addition, we carefully revised the tense issues and reduced the use of speculative terms in this version, hoping it can be more readable and precise.

**Point 7:** The authors should ensure that the data description fully complies with the journal's requirements. Additional details about the structure and usage of the dataset may be necessary for ESSD's standards.

**Response:**

Thanks for reminding. We added more details about the dataset in Lines 262-271:

The ASM-SS quasi-global storm surge dataset was generated from the ASM data-driven model established in section 3.3. The dataset is available at https://doi.org/10.5281/zenodo.13293595 (Yang et al., 2024a) as NetCDF files month by month from 1940 to 2020. Each file includes five parameters: longitude, latitude, nodes, time, and surge level. Longitude and latitude are the location information of nodes in degree; the unit of time is accumulated hours since 1900-01-01 00:00:00; surge levels are given in meters. Users can use longitude, latitude, and time as keywords to select surge levels at nodes of interest within a target period. In addition, the spatial resolution of nodes is 10 km along the coastline (as shown in Fig. 2). Since the sea surface varies rapidly during tropical cyclones, the temporal resolution of surge levels is set to hourly. Though this temporal resolution increases the data volume, it can provide sufficient information for users who want to analyze high-frequency variations of storm surges during extreme events.

**Minors:**

**Point 8:** The choice of an hourly temporal resolution for the dataset is not fully explained. The authors should provide a rationale for this decision, especially considering the implications for data volume and usability.

**Response:**

Thanks for your recommendation. The reason was added in Lines 268-271:

Since the sea surface varies rapidly during tropical cyclones, the temporal resolution of surge levels is set to hourly. Though this temporal resolution increases the data volume, it can provide sufficient information for users who want to analyze high-frequency variations of storm surges during extreme events.

**Point 9:** The mention of "small phase shifts" (line 120) lacks context. The origin of these phase shifts and their impact on the results should be discussed in detail.

**Response:**

Thanks for your suggestion. Previous research has discussed this issue. For example, Horsburgh and Wilson (2007) gave the following figure:

[Figure]

**Figure 4.** Schematic diagram of a sinusoid whose phase is altered but whose frequency and amplitude remain unaltered. The solid line (O) represents observations, the dotted line represents tidal predictions (T) and the dashed line represents the residual obtained via subtraction (R).

(Horsburgh, K. J. and Wilson, C.: Tide-surge interaction and its role in the distribution of surge residuals in the North Sea, J. Geophys. Res., 112, 2006JC004033, https://doi.org/10.1029/2006JC004033, 2007.)

In this version, we presented the reason in Lines 110-113:

(5) Finally, a 12-hour moving average was applied to SS data to limit possible remaining tidal signals (Tiggeloven et al., 2021; Yang et al., 2023), which are generally generated by small phase shifts in predicted tides due to the difficulty of obtaining perfect and completely accurate estimates through harmonic analysis (Horsburgh and Wilson, 2007).

**Point 10:** Units such as cm/m should be standardized across the manuscript. Similarly, decimal precision should be consistent for a more professional and coherent presentation of the data.

**Response:**

Thanks for your professional suggestion. We standardized the units of RMSE and MB to meters with three decimal places. The precision of CORR was set to two decimal places:

Lines 14-16: the precision of this model (medians of correlation coefficients, root mean square errors, and mean biases are 0.63, 0.093 m, and -0.049 m, respectively) is better than that of the state-of-the-art global hydrodynamic model (medians are 0.55, 0.106 m, and -0.044 m);

Lines 182-184: Figure 3(a-c) show that on a quasi-global scale (i.e., for ALL TGs), the median CORR of the entire time series of surges is 0.78, RMSE is 0.062m, and MB is 0.014m. In comparison, the reconstruction precision for extreme events (>95th percentile) is lower: CORR is 0.59, RMSE is 0.094m, and MB is -0.052m.

Lines 215-217: ASM (medians of CORRs, RMSEs, and MBs for 95th extremes are 0.63, 0.093 m, and -0.049 m, respectively) outperforms the numerical model GTSM (medians are 0.55, 0.106 m, and -0.044 m).

Lines 249-251: there are noticeable differences between ASM and GTSM. On the quasi-global scale, medians of CORRs, RMSEs, and MBs of the entire surges (95th extremes) between them are 0.32 (0.23), 0.084 m (0.138 m), and -0.056 m (-0.126 m),

**Point 11:** The gray lines in the figures (presumed to be tropical cyclone paths) should be explicitly described, and their inclusion justified. What purpose do these lines serve, and how do they enhance the understanding of the storm surge dataset?

**Response:**

Thanks for reminding. Since not all areas between 45°S to 45°N are affected by tropical cyclones (for example, the equatorial region, the South Atlantic and the southeastern Pacific), mapping the tropical cyclone paths can facilitate the division of sub-regions and highlight their differences. We added relevant descriptions in this version:

Lines 177-178: …Note that the equatorial region (~6°S to ~6°N) was separated as an independent area since it has almost no tropical or extratropical cyclones.

Lines 180-181: Figure 3: ASM model evaluation at tide gauges from 1940 to 2020. (a-c) Entire surge and 95th extreme evaluation statistics for different regions; (d-i) Distributions of evaluation metrics. Gray lines are tropical cyclone paths.

Lines 213-214: Figure 5: ASM model comparison with the numerical model at tide gauges from 1979 to 2018. (a-c) ASM and GTSM 95th extreme evaluation statistics for different regions; (d-i) Distributions of evaluation metrics. Gray lines are tropical cyclone paths.

Lines 245-247: Figure 7: Differences between ASM and GTSM at the coastal scale from 1979 to 2018. (a-c) Comparison statistics between ASM and GTSM modeled entire surges and 95th

extremes for different regions; (d-i) Distributions of comparison metrics. Gray lines are tropical cyclone paths.

**Point 12:** The same color bar is used for multiple metrics, which can create confusion. I recommend using separate color bars for each metric to avoid misinterpretation.
**Response:**

Thanks for your recommendation. We used three color bars to show different metrics in this version, hoping it can be clearer. Taking Figure 3 as an example:

[Figure]

Figure 3: ASM model evaluation at tide gauges from 1940 to 2020. (a-c) Entire surge and 95th extreme evaluation statistics for different regions; (d-i) Distributions of evaluation metrics. Gray lines are tropical cyclone paths.

**Point 13:** The use of "surge" as a variable name in the NetCDF files is problematic, as

it refers to a physical phenomenon rather than a dataset variable. I recommend choosing a more precise name that clearly describes the data field.

**Response:**

Thanks for your recommendation. We replaced it with 'surge level'. The new NetCDF files were updated in the repository.

**Point 14:** Line 62, a space between "abovementioned".

**Response:**

Thanks for reminding. This word was not used in this version since we rewrote the introduction.

**Another comment:**

**Point 15:** The manuscript emphasizes the computational inefficiency of numerical models, but fails to acknowledge that AI models, particularly those involving extensive preprocessing, ground truth acquisition, and training, can also be computationally expensive. Large/big AI models often require substantial computing power. A more balanced comparison of the computational demands of AI models versus numerical models would provide a fairer perspective on the advantages and limitations of each approach.

**Response:**

Thanks for your constructive suggestion. As you mentioned in **Point 1**, this paper should focus on the gaps between existing storm surge models or datasets to highlight the advantages of our dataset; another reviewer holds the same opinion. Computational efficiency is not the most concerning matter for dataset users. They care more about what the new dataset can provide to their research. Therefore, we rewrote the introduction and deleted relevant descriptions in other sections.

Nevertheless, as you point out, with the expanding application scenarios of AI, finding a more objective way to evaluate its computational efficiency and resource cost is worth attention.

**Summary:**

Overall, this manuscript presents a highly valuable and timely contribution to the field of storm surge modeling. The application of machine learning to generate a global, high-resolution dataset fills an important gap in coastal hazard prediction, especially for regions lacking sufficient observational data. The dataset's strong performance in

reconstructing extreme values, combined with its spatial resolution, demonstrates its potential for numerous applications in coastal risk management and scientific research. While there are areas that could benefit from further clarification and refinement, particularly in terms of methodological transparency and computational comparisons, the work is commendable. It reflects a significant step forward in leveraging AI for oceanographic data analysis, and with some improvements, it will undoubtedly become a highly valuable resource for the community.

---

## Author Response (AR2)

**Response to Reviewer#2 Comments**

The authors have addressed most of the concerns raised in the initial review, and the manuscript has improved significantly.

**Point 1:** However, regarding Point 2, while the authors added an extended explanation about the differences between single-site models and their proposed model, much of the text is verbose and does not add meaningful insights. It still fails to clarify what preprocessing or modeling steps make the model sensitive to geographic information. Additionally, the term "regional relationship" is unclear. That said, for a data-focused article, it is not strictly necessary to explain these points in detail. I recommend either deleting the redundant text (around line 140) or providing a simple, clear explanation. The manuscript can then be accepted.

**Response:**

Thanks for your constructive suggestion! In this version we deleted the redundant text and added a flowchart for the modeling processes of our model, hope this revision could be readable:

**2.5 All-site Modeling Framework**

Full details of the ASM can be found at Yang et al.(2023). Here, a brief description of its modeling processes is provided. Assuming there are six available TGs within 45°S to  $45^{\circ}N$  (Fig. 3(a)):

(1) Obtaining predictors (Fig. 3(b)). Four atmospheric data (mslp, u10, v10, and t2m) for each TG station are extracted from the ERA5 dataset through linear interpolation. Changes in sea level pressure and wind are the main factors in generating SSs (Woodworth et al., 2019); adding temperature variations considers the effects of thermal expansion and contraction. Meanwhile, following Yang et al.(2023) and Yang et al. (2024a), another three variables (longitude, latitude, and timestamp) are considered since geographical locations and record lengths of TGs are different. Hence, the predictor matrix for each TG consists of 7 columns: mslp, u10, v10, t2m, longitude, latitude, and time;

(2) All-site modeling (Fig. 3(c)). Predictor matrices and SSs of all six TG stations are stacked into one predictor matrix and one SS matrix. Then, the eXtreme Gradient Boosting Tree (XGBoost) (Chen & Guestrin, 2016) is used to learn the relationship between these two matrices. The XGBoost is a residual machine learning model that generates a new decision tree using SS residuals from the previous tree. Therefore, the new tree will pay more attention to training where the residual errors are significant, making it suitable for modeling SS extremes;

(3) Reconstruction (Fig. 3(d)). SSs can be estimated for any target node along the coastline by inputting the corresponding predictor matrix of that location into the model established in step (2).